# From Evidence Synthesis to Transfer: Results from a Qualitative Case Study with the Perspectives of Participants

**DOI:** 10.3390/ijerph19095650

**Published:** 2022-05-06

**Authors:** Cristina Lavareda Baixinho, Óscar Ferreira, Marcelo Medeiros, Ellen Synthia Fernandes de Oliveira

**Affiliations:** 1Nursing Research, Innovation and Development Centre of Lisbon (CIDNUR), Nursing School of Lisbon, 1900-160 Lisbon, Portugal; oferreira@esel.pt; 2Center for Innovative Care and Health Technology (ciTechCare), Polytechnic of Leiria, 2410-541 Leiria, Portugal; 3Nursing School, Federal University of Goiás, Goiânia 74690-900, Brazil; marcelo@ufg.br; 4Graduate Program in Collective Health, Federal University of Goiás, Goiânia 74690-900, Brazil; ellen@ufg.br

**Keywords:** clinical clerkship, evidence-based practice, knowledge management, learning, nursing, students

## Abstract

The increase in health research brings challenges to the production, synthesis, and use of research findings in clinical practice. In the case of undergraduate training in nursing, it is necessary to think about the curricular contents and create opportunities to develop skills for learning evidence-based practice. The objective of this study was to analyze nurses’ perspectives regarding the effects of their participation in a project of translation of knowledge into clinical practice during undergraduate nursing education, specifically involving knowledge, attitudes, and competencies related to the use of evidence. This is a qualitative case study grounded in the knowledge-to-action theoretical framework. The participants were 13 nurses who were involved in a project about the translation of knowledge into clinical practice during the last term of their undergraduate course. The data were collected by applying interviews between December 2020 and April 2021. Content analysis was carried out by using the qualitative data analysis software tool webQDA^®^. The following categories emerged from the content analysis carried out on the material gathered during the interviews: understanding evidence; learning how to use evidence; transferring evidence; adjusting to the context; and observing the advantages of evidence-based practice. Extracurricular activities were perceived as an opportunity to understand what evidence is and observe in loco the advantages of health care for clients, teams, and services. During the project, the participants developed cross-sectional competencies and envisaged changes to their professional activity as a result of changes in their attitude regarding evidence and its use. We concluded that the opportunity to develop evidence-related activities allows for the development of skills and influences the attitude towards evidence-based practice and knowledge use.

## 1. Introduction

There is a consensus that nurses must carry out evidence-based practice (EBP) in their professional activities and take leadership roles, in collaboration with other health professionals, to support reforms in health care that include, among other factors, the introduction of research results in practice contexts so that the health care offered to the population can be improved [1,2,3]. The importance of EBP has been corroborated by several authors [1,2,3,4,5]. However, many barriers to the “universalization” of its systematic use in clinical practice have been identified: a lack of knowledge and competencies related to evidence use, synthesis, and implementation, and the low scientific literacy of nurses [2,6]. Some authors have stated that, although many educational strategies have been applied and evaluated with students and professionals, a lack of knowledge and competencies related to EBP can still be found in different organizations and is an obstacle to its use [3,6].

Other barriers to implementing EBP include difficulties related to the time and resources necessary for training in this area, a lack or insufficiency of academic programs on the subject, inconsistencies between learning and practice environments chosen by students, logistic challenges related to introducing the development of competencies in a system traditionally based on learning a specific set of knowledge [4], and a lack of confidence [7], since EBP requires proficiency in applying knowledge [1].

The content about EBP is integrated into the syllabuses of health programs in undergraduate and graduate nursing courses and continuing education initiatives [3,6]. However, it is not always evaluated with a level of methodological rigor that allows the development of knowledge about the efficacy of the approach and a deep understanding of how this content really takes effect and what processes contribute to intended and unintended results [4]. This makes it impossible to provide guidance for academic and clinical educators regarding the interventions that are most effective in improving the quality of education addressing evidence [5].

The interest in this phenomenon has been increasing over the past decade with recommendations that future health professionals be involved in this type of activity in order to support a culture and spirit of research [8,9]. This movement has been occurring simultaneously with some alterations in nursing course syllabuses to deepen theoretical content [10]. However, few programs promote the integration between theory and practice, especially by means of research opportunities and/or the application of evidence during clinical clerkships that support the adoption of professional practice guided by research results [6,11].

Health care is becoming increasingly complex and now requires more scientific competencies [9], but a literature review has showed that the traditional educational model, in which professors prepare classes, explain, and demonstrate to students is not enough to train professionals whose practice is supported by evidence [12], if the expectation that they know how to use scientific studies to produce safe results in their clinical practice is considered [6]. The great challenge to education in nursing is to allow students to gain EBP-related competencies and strengthen their ability to think, solve problems, and develop clinical reasoning [9].

In Portugal, the nursing degree has about 50% of the contact hours for clinical training in different settings so that students can develop clinical skills for the provision of quality nursing care that ensure people’s safety. So, this type of teaching assumes a vital importance mainly because of the associated learning opportunities and the theoretical-practical integration of knowledge [11], creating an opportunity for the integration of theoretical knowledge about research and evidence and simultaneously enabling the practice of activities related to these.

Faced with the above our research question emerged: does the participation of nursing students in knowledge translation activities during clinical teaching enable the acquisition of knowledge and the development of attitudes and skills for the use of evidence in clinical practice?

The objective of the present study was to analyze the perspectives of nurses regarding the effects of their participation in a project of translation of knowledge into clinical practice during their undergraduate nursing course, specifically involving knowledge, attitudes, and competencies related to the use of evidence.

## 2. Materials and Methods

### 2.1. Study Design

The study was a qualitative case study [13] grounded in the knowledge-to-action theoretical framework [14]. The model, which was developed by Graham et al., defines two cycles that are pivotal for knowledge translation: knowledge creation and action, which represent the process of knowledge application [14]. In the graphic representation of the model, the knowledge creation cycle is presented as its central nucleus, and oriented so knowledge that is perceived as useful can be applied in clinical practice. To achieve this goal, this cycle includes the synthesis and creation of products and/or instruments that promote the introduction of knowledge into clinical practice [14]. The action cycle includes seven dynamic interconnected practices: (a) selecting, adjusting, and implementing interventions; (b) evaluating factors that hinder or facilitate the use of knowledge; (c) adjusting knowledge to the social context; (d) identifying the problem; (e) continuing to use knowledge; (f) evaluating results; and (g) monitoring the use of knowledge [14].

### 2.2. Participants and Setting

The participants in the present study were nurses who participated in a research project to get their degree; it was entitled “Safe Transition,” and took place during the last term of their undergraduate nursing course. This project involved three institutions that developed activities in partnership: a nursing school, a hospital, and a set of primary healthcare institutions in Lisbon and the Tejo Valley region, Portugal. The specific purpose of the project was translating knowledge into clinical practice. The nurses participated in the project during their clinical clerkship, which was carried out in the last term of the course and aimed to integrate them into professional life, as an extracurricular activity. Participation in the project was simultaneous with the clerkship, for which there were predefined activities and schedules. The participation of the undergraduates was voluntary and followed their signature of free and informed consent forms. The activities in which the students would participate under the guidance and supervision of professors and clinical supervisors were previously defined by the nursing school and the service institutions.

The choice of participants was intentional, as we intended to study how participation in this activity led to knowledge, attitudes, and competencies related to the use of evidence. The inclusion criteria for the participants were: having participated in the project “Safe Transition” during the last term of their undergraduate nursing course; being involved in knowledge creation and/or knowledge application activities; having over 12 months of professional experience; and not working at the institution where they did their clinical clerkship. The exclusion criteria were: having less than 12 months of professional experience and working at the institution where they did their clinical clerkship. Thirteen out of 15 nurses who were eligible to be included in the sample of the present study agreed to participate in it.

The number of interviews made theoretical saturation possible, the sample size of qualitative studies being debatable, as well as the possibility of saturation, in the case of this research, from the 10th interview no new categories emerged.

### 2.3. Data Collection

“Safe Transition” is an intervention project. The choice of a qualitative study, which is used to understand people’s beliefs, experiences, attitudes, behaviors, and interactions [13], was based on the perception that integrating qualitative research into interventional studies is a strategy that has received attention in all subjects, given the ability of this research method to add a new dimension to interventional studies that cannot be obtained by simply measuring variables [13]. This approach provides science with the understanding of the experience and involvement of study participants [13].

The data were collected by applying semi-structured interviews between December 2020 and April 2021. The interview script was designed according to the knowledge-to-action model [14].

The stimulus questions for the interview were: 1. Do you consider that the involvement in the project allowed you to develop knowledge about EBP practice? 2. Do you consider that the participation in the project changed your attitudes towards the use of evidence in clinical practice? 3. Tell me a little about the skills that you have developed through your participation in the project; and 4. Which of the acquired skills have been the most mobilized/useful during the first years of professional practice?

The interview was conducted by a senior researcher from the team, who knows the context and the safe transition project, but was not involved in the clinical projects that the participants were part of.

### 2.4. Data Analysis

The online interviews, which were carried out by using the Colibri^®^ platform, were recorded and transcribed by the researcher who conducted them. Subsequently, they were independently analyzed by two researchers, according to the content analysis technique [15] and by using the qualitative data analysis software tool webQDA^®^. This facilitated the organization of the results of the interviews, analysis of the information, collaborative work, and validation of the categories by the research team.

The process of codification and validation of the results involved transcription, reading, codification, the definition of categories, and the return of the interpretation to the participants for it to be validated, in accordance with the theoretical framework [16]. 

The researcher who made the interview and transcribed it carried out the coding of the free codes. Another research using the software tool carried out the analysis of the findings independently, after which the team met to validate the content analysis carried out.

The analysis carried out by the researchers was emailed to the participants who were asked to read and comment on the categories and subcategories and whether they considered the interpretations reflected their experiences/ideas/feelings. This validation of data by the participants reinforces the external validity and saturation of the findings. 

### 2.5. Ethical Aspects

Ethical and formal principles were observed, from authorization for the students to participate in the project as an extracurricular activity to the right to not participate. The proposal was approved by the hospital’s research ethics committee as per report 09/2019 HVFX. 

The responsible investigator was responsible for managing all the data collected, screening those that could be shared with the team and those that remained with restricted access. At no time were used sociodemographic variables that could identify the different participants in the study.

The principal investigator had the responsibility to assign a unique code to each participant and only they knew the correspondence between the code and the participant’s personal data, with each professional being identified by a code made up of the letter N (for “nurse”) and a number (1, 2, 3, …) to guarantee participants’ anonymity and data confidentiality.

All documents (electronic and physical) have remained archived, in security, with restricted access, since the conclusion of this study, for the time defined by law. The list of the coding of the participants was maintained only by the investigator, not allowing access to other persons.

## 3. Results

The participants of the present study were 13 nurses who had been included in a project oriented toward transferring knowledge into clinical practice that was carried out during their last clinical clerkship in the undergraduate nursing course. The participants, of whom nine were women, had an average age of 26.6 (±3.14) years and had worked as nurses for an average of 57.08 (±43.6) months since graduation, with a minimum and maximum time of 18 months and 10 years, respectively. The average duration of the interviews was 40.44 (±12.99) minutes.

The following categories emerged from the content analysis: understanding evidence (register units (RU) = 62); learning how to use evidence (RU = 91); transferring evidence (RU = 63); adjusting to the context (RU = 21); and observing the advantages of EBP (RU = 57) (Table 1).

### 3.1. Understanding Evidence

Regarding the first category, the participants emphasized the importance of their participation in the project for their understanding of what EBP is, because, as expressed by one participant, “*It helped us perceive what Evidence Based Practice is*, […], *it is a bit similar to what they say in nursing school, but when we do our clinical clerkship we focus on reproducing what our supervisors do. Therefore, during the project, it was clear that evidence is in papers, in scientific databases, and it is not always put into practice*, […], *it is hard to make it get to services, because people are not always studying and looking for it*” (N01). Regarding integration between practice and theory of knowledge about evidence, one participant said that he “*had had contact with and discussed the expression Evidence Based Practice during theoretical classes, but it was during the course of the project that it was possible to understand its true meaning and develop competencies to support practice with evidence*” (N07).

The participants also mentioned that clinical clerkships do not always provide the opportunity to compare the ways to do what is recommended by research results. One participant said it was because “*When we start working or doing our clerkship, there is a bit of a tendency*, […], *not to guide ourselves by evidence, and we do things based on what the supervisor nurse does or says or what you hear other nurses say. From really simple things, such as application of ointment on the skin, dilution of antibiotics, or preparation for patient discharge, simple stuff, to shift organization regarding records, lots of things, really lots of things!*” (N11).

The importance of thinking about EBP for a change of attitude as undergraduates when faced with evidence presented in scientific publications emerged, together with the perception of the EBP concept: “*Even because the experience was enriching in terms of helping us avoid automatic practice, that is, helping us, as undergraduates, think about what we do, how we do it, and based on what rather than encouraging us to carry out practice by imitation, always doing the same, without thinking if there are already other solutions. It is necessary to raise awareness, for instance, of the main automatic and standardized practices, and promote reflective practice. But it is also necessary to allow professionals and students to have access to the most reliable databases. We need to invest in training of who has the power of defining strategies, developing actions, and inspiring future care providers*” (N13). “*It even influenced me to realize that we often use evidence poorly, not really use it poorly, we do not use it at all, we say that we do everything based on knowledge… which may even be true, but it is not the most recent knowledge*” (N04).

Understanding what EBP is and analyzing its advantages, but also reflecting on the way nurses work, make decisions, and are influenced by culture and shaping in clinical contexts are fundamental steps for professionals to adjust their practice so that it is aligned with knowledge and critically analyze their performance as a way to develop professionally.

### 3.2. Learning How to Use Evidence

The category learning how to use evidence was the one with the highest relevance regarding the record units. One nurse declared that “*During the undergraduate course, when we are students, we neither know how these activities are carried out nor have the chance to carry them out, which is why the clerkship is a great opportunity to learn how to formulate projects and develop activities that change practices*” (N03). Another pointed out that the project was important “*because there are gains regarding knowledge*”, and that when information about EBP was acquired, it also allowed nurses to develop “*important competencies for searching for and identifying needs and problems*” (N02). More specifically, one participant said that this included “*identifying needs and problems; formulating questions; planning, implementing, and evaluating interventions; identifying the best available evidence; and developing synthesis/summarization, critical thinking, and decision-making skills*” (N13).

Learning about evidence is a process that occurs at different levels by means of research activities, critical reading of studies, evidence synthesis, and opportunities to carry out research activities in ongoing research processes, as reported by one of the participants: “*We force ourselves to search about the subject, see things that are already being done, to be able to better justify our practice; The research part also helps, because it improves our language a little, that is, it makes us use more scientific language, not just scientific, but also technical-scientific terms; in other words, helps us to truly understand what a clinical trial or validation is, and when we use the terms we know exactly what they are about* […]. *I have a better notion of what data collection is, of how to consult databases, and of data analysis, and I have a very clear idea of how they are related*” (N06).

Learning about evidence is achieved by integrating theoretical knowledge into practice. However, extracurricular activities allow students to obtain new knowledge by means of participation in activities that offer the possibility of identifying clinical areas that are a priority for the introduction of evidence, continuing to use knowledge, evaluating results, and monitoring the use of this knowledge.

### 3.3. Transferring Evidence

Regarding the category transferring evidence, one participant said that “*Evidence Based Practice also allowed us to perceive how things are changed*” (N09). Aspects related to the process of the introduction of research results and barriers to it emerged in the interviews.

One of the nurses declared: “*I found myself talking to the articles and thinking ‘so this intervention is for solving the problem straightaway, so this is the path I want to take; how am I going to prioritize it? I have my shift ahead; is it possible to do something different already?’*” (N02). Another participant mentioned that “*We knew, in theory, that we would be able to improve the outcomes of the patients if the knowledge were transferred into practice. Then, once we got to the practice part and evaluated everything*, […], *we ended up realizing that sticking to studies really makes sense. I mean, putting them into practice and producing more studies, which is what we ended up doing after coming across evidence and seeing how to put it into place so it could provide results, because we saw all this in the project. Therefore, learning this theoretical framework, applying it, and evaluating it ended up helping me realize the importance of Evidence Based Practice*” (N10).

In contrast, the experience even allowed “*the confirmation of what people say about the difficulties of using the results of scientific research*” (N04). The identified barriers were related to personal, temporal, and organizational factors. According to the participants, “*Obstacles are inevitable. Personal and professional experience and routine procedures are the greatest barriers in my opinion, but there is also the issue of constant updating in the health area, with the need for massive personal investment that often takes a heavy toll. This leads to large gaps between currently recommended practice and care that is actually delivered*” (N13). “*Another thing we say a lot, I and my colleagues, those who were with me, is that it is a long process and it is not easy to introduce this information into services, because each person has their own ideas, their own way to do things and, even with training, which not everybody attends, it is not possible to change things quickly*” (N01).

Transferring knowledge into clinical practice was achieved by resorting to different strategies, and an analysis of the barriers allowed the implementation of measures to control them and make changes possible. Negotiation in the teams and the definition of strategies between the people involved in the project (including the students) promoted collaborative work, problem solving, and the introduction of evidence into clinical practice.

The methodology used allowed the participants to learn how to adjust evidence to the context, which was illustrated by the following accounts: “*I remember that the professor raised the question of this difficulty when we first got involved and she warned us that, sometimes, it is necessary to adjust knowledge to the reality of a given context, because studies can be carried out in other countries. Then we are sort of importing results into our reality, which may be different, even because of how our care procedures are organized*” (N01). “*We did not have the necessary resources that other places would have and we gradually adapted, with the help of the professor and the nurse who were there. We identified what we could do with the materials, team, and physical space we had and, after reading the systematic literature review and getting to know the project that also existed*, […], *we ended up realizing that we had some limitations and would have to adapt because of them and work with what was possible*” (N10).

### 3.4. Adjusting to the Context

Adapting to the context is necessary because of the available material and human resources, but also because of the characteristics of the service, organization of care procedures, and dialogue between what exists in clinical practice and what is recommended by evidence. “*The truth is that we already had knowledge about what a cerebrovascular accident is and care that should be delivered, but we were there, so we had to be good at care delivery. Consequently, I had to have this knowledge, and being able to verify whether to do something or not in the context itself really makes sense. That is, it makes sense that work is* in loco *and oriented toward the needs of the context. Being* in loco *is a fully feasible strategy because I was doing that search* […], *I was looking for evidence to guarantee that care would be guided by the improvement of the patient, which makes total sense to me. I was, so to speak, like a means to empower that person, and that motivated me, because I knew that my search process would result in gains. Gains for the patient, and this motivated me more to read, and search, and identify what was necessary*” (N11).

The analysis of the context showed that the opportunity to adjust evidence to the context at issue was positively regarded by the participants, not just because of the lesson of how to do something, but also because the comparison between evidence and what is actually carried out allows them to think about practices and advantages of using evidence, which feeds back into the process that leads to a transfer of knowledge into practice, so its applicability is guaranteed.

### 3.5. Observing the Advantages of Evidence-Based Practice

In the category observing the advantages of EBP, there were accounts about the results obtained in the project and the recognition of how it helped improve care quality: “*The project also ended up hugely raising awareness of that and about what is necessary, the conditions of the patient at hospital discharge, what they need regarding nursing care. That is because most of the time the patient is discharged but is not trained to carry out activities of daily living, is dependent regarding self-care, cannot manage medications yet, and cannot manage post-discharge everyday life, and there was no time to inform them, provide education about health, train them to do what they have to at home*” (N05). “*It was also important to realize that people recognized the importance of our work and how it would improve the preparation for hospital discharge for patients and relatives, and it would even impact their return home, because people became more independent*” (N03).

The participants stressed the importance of verifying these advantages for their own motivation: “*We had already carried out searches for other college projects, but not for implementation with users, in the service, and improving care delivered to patients, not for implementing things and seeing our project being developed and bearing fruit. We realized that it was really important*” (N07).

In fact, the project had indicators that were measured every six months, which allowed an assessment of its results and the use of EBP. The improvement in the indicators related to care delivery encouraged professionals to keep using EBP and monitor its results.

## 4. Discussion

The views of the 13 nurses who participated in the present qualitative study corroborated the results of other studies that have reported that education in EBP does not occur in schools [10,17]. Instead, it relies on clinical experience, contact with contexts that really involve the use of EBP to support clinical decision-making based on knowledge, and opportunities to participate in research projects and get to know research methods and techniques [11]. Therefore, most undergraduate nursing students who complete the syllabus units related to scientific research are neither effectively qualified regarding research methodology and the execution of research processes [18] nor fit to adopt actual EBP [19,20].

Gaps in knowledge and competencies are some of the factors that lead clinical nurses to apply a range of decision-making processes during their practice that are not always supported by evidence [7,11,19]. Consequently, professionals favor, for instance, the observation of the work carried out by more experienced nurses and the information these veterans provide for their own decision-making [19]. This finding reiterates the need to encourage a discussion, both among professionals and in the academic community, about the models used to teach about research, use of evidence, and decision-making based on it [6,9,10,11].

The findings of the present study reinforced the results of an integrative literature review on the attitude of undergraduate nursing students regarding research. It showed that taking a research-related subject, having interest in a specific research field in nursing, and having the possibility to apply research knowledge and previous research experiences (including data collection and analysis) allowed for marked improvement in the attitudes of the students regarding research and its use [8]. The accounts of the participants clearly indicated that the extracurricular activity during their course allowed them to understand what evidence is, learn how to read and analyze it, and transfer it into clinical practice.

Studies on the subject have shown the advantages of EBP but have not conclusively addressed the impact of these experiences once they are completed and what strategies should be adopted to effectively use EBP at the beginning of a career. A program oriented toward initiating students in research suggested that active and early participation of nursing undergraduates in research activities encourages new graduates to incorporate evidence into care delivery and seek a graduate degree [17]. In another study in which the participants were former students with at least 18 months of professional experience, record units related to understanding what evidence is, learning how to use and transfer evidence, and observing advantages of EBP emerged in the interviews. This seems to point to an influence on its contribution to education and the development of competencies related to the action cycle for the process of knowledge application [14]. Future studies could explore the influence of this type of activity on the adoption of EBP in the first years of professional experience. It is important to identify what strategies should be used, what paths should be taken, what plans should be implemented, and what corrective evaluations should be carried out so that, in the future, the practice of nurses who have just entered the profession will truly be based on evidence rather than obsolete and outdated knowledge.

A study on the use of EBP reinforced the gains in knowledge and motivation [20]. In addition to the advantages in the education area, EBP has the potential to help improve the quality and results of health care [21], which was appreciated by the participants. 

The results also identified difficulties and barriers in the implementation of knowledge in clinical settings, which corroborates the results of other studies whose conclusion was that, despite the individual efforts of nurses, it is difficult to create an organizational culture that can support EBP and make it feasible. Some studies have recommended that leaders try to minimize expected barriers and base institutional policies on evidence [22,23,24,25,26]. Concomitantly, the academic community should find support for EBP-related education in clinical practice contexts, in which nursing care, inserted into an interprofessional cooperation network, has become increasingly complex [24] and discernible.

Therefore, it is necessary to design more studies on the challenges of transferring knowledge to final consumers (professionals) and beneficiaries (care users) and the way it is carried out, which would bridge the conceptual and pragmatic gap between what is produced and what is used to actually improve health. The creation of communication and collaboration networks involving researchers, professionals, and citizenships would be useful in this process [27].

### Implications and Limitations of the Study

In view of the results, we recommend that clinical teaching students have the opportunity to use research results and learn to transfer them to clinical settings because we believe that this will be a core competence for the health professionals of the future, with an impact on improving care and the sustainability of health systems. The international nursing movement should discuss the need to include in curricula and clinical learning contents and activities related to evidence synthesis and its transfer to clinical settings.

This study had limitations regarding its method, data collection technique, and data analysis. The intentional choice of participants and the concrete experience of having participated in the safe transition project limited the results to the context. The interaction between researcher and participant may have influenced the response to what is socially accepted.

Despite the limitations, the results provided new knowledge about the contribution of the participation of students in knowledge transfer projects during the last clinical clerkship in undergraduate nursing courses. This step presumably would integrate students into professional life, for their acquisition of knowledge and development of competencies to be applied to EBP.

## 5. Conclusions

The 13 nurses who participated in the present qualitative study, which was grounded in the theoretical framework of the knowledge-to-action model cycle, participated in a project oriented toward transferring knowledge into clinical practice during the last clinical clerkship of their undergraduate course. The following categories emerged in the content analysis: understanding evidence, learning how to use evidence, transferring evidence, adjusting to the context, and observing advantages of evidence-based practice.

The findings showed the effect of this participation on the understanding of what EBP is, decision-making based on knowledge, and critical analysis of their professional performance. Education about evidence was achieved by integrating theoretical knowledge into clinical practice, which allowed for the verification, in a clinical context, of the advantages of this learning process. It also involved gains for users and improvement of indicators established for evaluating the project. The opportunity to verify gains in loco helped support the use of knowledge and encouraged this practice.

Future studies could explore the results of other educational programs addressing evidence, in coordination with clinical services, and examine the relationship between attitudes and competencies acquired by students during their academic trajectory, and the adoption of actual EBP during their professional activities.

## Figures and Tables

**Table 1 ijerph-19-05650-t001:** Corpus of content analysis. Lisbon; Portugal.

Category	Subcategory	Register Unit
Understanding evidence	Understanding what evidence is	31
	Reflect on the advantages of using evidence	16
	Link theory to practice	15
	**Subtotal**	**62**
Learning how to use evidence	Search science	28
	Assessing the quality of articles	25
	Interpreting results	19
	Communicate science	19
	**Subtotal**	**91**
Transferring evidence	Transferring evidence—process	38
	Products and strategies for introducing evidence	20
	Transferring evidence—difficulty	5
	Subtotal	63
Adjusting to the context	Analysis of the context	13
	Dialogue between theory and practice	8
	**Subtotal**	**21**
Observing the advantages of EBP	Improving quality of nursing intervention	27
	Advantages to patients’ clinical outcomes	26
	Communicate science to patients	4
	**Subtotal**	**57**

## Data Availability

The data used during this study are available from the corresponding author, upon request by e-mail.

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
