# Peer review of "From Evidence Synthesis to Transfer: Results from a Qualitative Case Study with the Perspectives of Participants"

_ijerph, 2022, doi:10.3390/ijerph19095650_

Round 1

Reviewer 1 Report

Thank you for having a chance to review the interesting and valuable manuscript entitled “From Evidence Synthesis to Transfer: Perspectives of Participants of a Nursing Undergraduate Project”. This study will make a contribution toward the improvement of nursing education.

Introduction: I think you had better emphasize the differentiation of this research.

Materials and Methods: 2.2. Study Location and Participants: I would like to suggest you to clearly present the location (which province, which country).

Results (line 152-290): I would like to suggest you to use subheadings of the categories emerged from content analysis for readability.

Results (line 219-229, 287-290): I think you had better write these phrases in italic.

Discussion: I would like to suggest you to present theoretical and practical implications of this study.

Author Response

Dear reviewer:
We thank you for your attention to the review of our article and recommendations made that we have introduced and that have helped to improve the quality of the article.
We have shaded the changes made in yellow to facilitate proofreading.

Reviewer 2 Report

Thank you for giving me to review your manuscript. This manuscript is interesting for considering nursing education and how to progress with educational systems. However, the following points should be critical for the quality of qualitative research.

The title should contain information on the research design.

The abstract should contain the background information on this topic.

In the background, the authors should be conscious of paragraph writing. 

The background should focus on nursing education regarding evidence-based education and competency. The present content is too broad.

The authors should describe the evidence gap concretely regarding nursing education by referring to a wide range of research from developed and developing countries. The following articles could enrich this research background.
-Pickett, S., Options for Teaching Physical Assessment Skills On-Line for Nurse Education Students. Teaching and Learning in Nursing, 2017. 12(1): p. 32-34.
-Douglas, C., C. Windsor, and P. Lewis, Too much knowledge for a nurse? Use of physical assessment by final-semester nursing students. Nurs Health Sci, 2015. 17(4): p. 492-9.
-Ohta, R., S. Maejma, and C. Sano, Nurses' Contributions in Rural Family Medicine Education: A Mixed-Method Approach. Int J Environ Res Public Health, 2022. 19(5).
-Maejima, S., R. Ohta, and C. Sano, The Implementation of a Clinical Ladder in Rural Japanese Nursing Education: Effectiveness and Challenges. Healthcare (Basel), 2021. 9(4).

The background should contain research questions.

The authors should describe what kinds of methods they used in the section on participants. 

The authors should validate the number of participants.

The authors should add the interview guide to the semi-structured interviews.

The authors should describe the presence of the theoretical saturation and member checking clearly for the research credibility in the analysis section. 

The discussion should summarize this research's results and significant points by revising the previous issues.

Author Response

(The authors gave the same response as above.)

Reviewer 3 Report

Dear authors/s,

I would like to thank you for the opportunity to review the manuscript you submitted to IJERPH. The topic is very interesting and offers an insight into the perspectives and experiences of participants on knowledge translation from nursing educational settings to clinical practice. Despite being an interesting reading it needs some more polishing. Here is a list of my observations which I hope you will find useful:

  • The title of the manuscript needs more work, as it could be more focused. The concept of knowledge translation should be very clear. It would also be advisable to point to the research design used within the title.
  • I would suggest in the abstract referring to the sample size in the methods section and not in line 23 when discussing results. The abstract is missing a clear conclusion.
  • There is a logical flow in the introduction section, however, this part could be more comprehensive as it is written from a more general perspective, putting less weight on the nursing perspective. The majority of literature used in this part (despite being limited) should be up-to-date.
  • Clearly define the type of qualitative research design.
  • The Sections 2.2 and 2.3 should be one section Participants and setting. Determine how did you reach the criterion of saturation or criterion of information power if you predetermined that your sample consists of 13 participants?
  • The tile of the section Data treatment is not appropriate. In this section, mention was done in order to secure the trustworthiness and credibility of the results.
  • As a reader, I would expect to have the steps of data analysis written in a more rigorous way.
  • Section 2.6 – describe the type of informed consent, describe where the data are kept and for how long, possible remunerations,
  • Section 2.4. – the interview guide is missing or should be attached as a supplement file.
  • The results section is clearly missing a figure or a table of the categories and sub-categories or codes that were identified. Please see samples of qualitative articles.
  • Page 4, line 9: “The following categories emerged from content analysis: understanding evidence (n=62); learning how to use evidence (n=91); transferring evidence (n=63); adjusting to the context (n=21); and observing the advantages of EBP (n=57).” What does this number mean? It is highly unusual in qualitative methodology to list quantitative data this way. What does this mean? Similar to one paragraph before this – the description of the sample (in qualitative studies SD is not important as the sample size is purposive and the nature/philosophy behind the paradigm is quite different). I would suggest to the reader or use a sample of a qualitative article to see who to write this part.
  • Did the participants use abbreviations in the interview EBP or did you insert them? The rules of transcription are very clear. I suggest that they are being followed.
  • The quotations in the results section are somewhat long. I suggest a revision and more focus on the key quotations.
  • Limitations are poorly addressed within the article. This needs to be done.
  • Similar to in the introduction, the discussion needs more focus on nursing, as was promised in the title.

Author Response

(The authors gave the same response as above.)

Round 2

Reviewer 2 Report

The manuscript has been considerably improved. I think that this paper is suited for inclusion in our journal.

Author Response

Dear review:

Thank you for your attention to our article.

Best regards;

Reviewer 3 Report

You significantly improved your manuscript. Congratulations.

Author Response

Dear review:

Thank you for your attention to our article.

This manuscript is a resubmission of an earlier submission. The following is a list of the peer review reports and author responses from that submission.

Round 1

Reviewer 1 Report

This article From evidence synthesis to transfer: perspectives of participants of a nursing undergraduate project presents a qualitative study exploring the perspectives of 13 nurses regarding the impact of their participation in a knowledge translation project during their education, on the use of evidence into clinical practice. The knowledge translation project took place during the last clinical placement and was optional. Participants in the qualitative study were recruited after at least one year of clinical practice and participated to an interview. A thematic analysis was performed and 5 main themes were identified: understanding evidence ; learning how to use evidence ; transferring evidence ; adjusting to the context ; observing the advantages of EBP.

The topic could be of interest for undergraduate program coordinators or educators in these programs. However, the study shows several important weaknesses and methodological flaws. Major points are the following :

  1. I understood that the objective is to explore the perspectives of nurses regarding the impact of their participation in a knowledge translation project during their education, on the use of evidence into clinical practice. But this is not clearly stated in the manuscript. The objective should be specified.
  2. The KTA framework is mentioned as the underpinning framework (line 78: “Qualitative study grounded in the Knowledge-to-Action theoretical framework”). However the use of this framework in the study is insufficiently described (if possible the interview guide could be provided as supplemental data). It is said to be used to build the interview guide. The KTA could have also been used to structure the analysis but it is not described. Furthermore I would propose not to use the wording “grounded in the KTA theoretical framework” (line 78) as the KTA model is not a qualitative approach or theory. KTA could be mentioned for example as a model used to developed the interviews or to guide deductive analysis but not as mentioned in the manuscript line 78.
  3. In the KTA action cycle description (lines 85-89), the “seven interconnected practices” seems not to be exactly what is described in the model. To my opinion, even if it is an iterative process, a presentation in the order presented by Graham et al. and with the same wording would help to better understand the different steps of the action cycle.
  4. The education project (safe transition) should be described in more details (or a reference to an earlier publication could be provided). It is sometimes defined as a research project and sometimes as an implementation project.
  5. Reporting of qualitative study requires authors to precise several aspects that are not presented in the current manuscript. This concern in particular: the clarification of the methodological orientation and the description of the interviewers and their relation with the participants. I recommend that the authors use of one reporting guidelines for qualitative research : COREQ (https://pubmed.ncbi.nlm.nih.gov/17872937/) or SRQR (https://pubmed.ncbi.nlm.nih.gov/24979285/) for example to verify if every needed information is presented.
  6. Sampling and external validity : the authors mention the only 15 nurses were eligible to participate. It is also mentioned that on of them has 10 years of experience. So it seems that the program concerns only a very small proportion of the students. This is an important point that need to be clarified and discussed in the article. And if the program cannot be offered to each student in a program then the interest of the presented study for program coordinators is clearly diminished. In the same order of ideas, the fact that only volunteer students participate and that the education project is extracurricular should be discussed in the manuscript. And finally, the sample interviewed seems very heterogenous in terms of year of practice. This should be discussed. Indeed, the impact (and the extend to which participants remember it) of a program involvement is not the same after 1 or 10 years of activities.
  7. Inclusion criteria: please explain the inclusion criteria: “being involved in knowledge creation and/or knowledge application activities” (line 112-113): justification and clarification of what is meant here.
  8. A part of the justification of the use of a qualitative method to gain knowledge on participants experience is not necessary and add confusion (line 122-124). I propose to delete the following part: “was based on the perception that integrating qualitative research into intervention studies is a strategy that has received attention in all subjects, given the ability of this research method to add a new dimension to intervention studies that cannot be obtained by simply measuring variables.” And to rephrase the rest of the justification text.
  9. Qualitative analysis: the authors cite the “content analysis technique” (line 136) but it seems that the description of the process (lines 139-141) and the way they present the results (but the n= of citation in each category – lines 157-159), is rather thematic analysis (Braun & Clarke 2006). This should be clarified.
  10. The presentation of the results is too long and the organization by themes or category is not clearly highlighted. Citations are not all iconic and should be shortened and further selected.
  11. The discussion should begin with the main results.
  12. The exposed limitations (lines 379-380) are too general and should be specified.
  13. Line 393: “The findings showed the effect of this participation…” goes behind the data. This should be revised.

Minor modifications:

  1. B-learning modality (line 108): should be explained
  2. The n= (lines 157-159) should be explained (or deleted if thematic analysis was performed).
  3. Exclusion criteria (lines 115-117): these should not be the opposite of the inclusion criteria. I recommend to delete the repetition.
  4. Ethical aspects (lines 143-144): the participation in the project should refer to the participation of the qualitative study in this chapter but it seems to refer to the education project. This should be clarified.
  5. I am not a native English speaker so I can only partly judge English but it seems to me that sentences are too long and I had difficulties in understanding some sentences in the discussion (e.g. lines 340-343 (In another study…).

Reviewer 2 Report

The 13 nurses who participated in the present qualitative study, which was 386 grounded in the theoretical framework of the Knowledge-to-Action model cycle, partic- 387 ipated in a project oriented toward transferring knowledge into clinical practice during 388 the last clinical clerkship of their undergraduate course. The following categories 389 emerged in content analysis: understanding evidence; learning how to use evidence; 390 transferring evidence; adjusting to the context; and observing advantages of evi- 391 dence-based practice. 392

The findings showed the effect of this participation on the understanding of what 393 EBP is, decision-making based on knowledge, and critical analysis of their professional 394 performance. Education about evidence was achieved by integrating theoretical 395 knowledge into clinical practice, which allowed for verification, in a clinical context, of 396 the advantages of this learning process. It also involved gains for users and improvement 397 of indicators established for evaluating the project. The opportunity to verify gains in loco 398 helps support the use of knowledge and encourages this practice. 399

Future studies could explore the results of other educational programs addressing 400 evidence, in coordination with clinical services, and examine the relationship between 401 attitudes and competencies acquired by students during their academic trajectory, and 402 the adoption of actual EBP during their professional activities.